# Re-Hamiltonian Generative Networks

## Reproducibility Summary

**Scope of Reproducibility**

The main objective of the paper is to "learn the Hamiltonian dynamics of simple physical systems from high-dimensional observations without restrictive domain assumptions". To do so, the authors train a generative model that reconstructs an inputted sequence of images of the evolution of some physical system. For instance, they learn the dynamics of a pendulum, a body-spring system, and 2,3-bodies. In addition to these environments, we further expand the testing on two new environments and we explore architecture tweaks looking for performance gains.

**Methodology**

We implement the project with Python using Pytorch [11] as a deep learning library. Previous to ours, there was no public implementation of this work. Thus, we had to write the code of the simulated environments, the deep models, and the training process. The code can be found in this repository: https://github.com/CampusAI/Hamiltonian-Generative-Networks A single training takes around 4 hours and 1910MB of GPU memory (NVIDIA GeForce RTX2080Ti).

**Results**

We found the model's input-output data slightly unclear in the original paper. First, it seems that the model reconstructs the same sequence that has been inputted. Nevertheless, further discussion with the authors seems to indicate that they input the first few frames to the network and reconstructed the rest of the rollout. We test both approaches and analyze the results. We generally obtain comparable results to those of the original authors when just reconstructing the input sequence (30% average absolute relative error w.r.t. to their reported values) and worse results when trying to reconstruct unseen frames (107% error). In this report, we include our intuition on possible reasons that would explain these observations.

**What was easy**

The architecture of the model and training procedure was easy to understand from the paper. Besides, creating simulation environments similar to those of the original authors was also straightforward.

**What was difficult**

While the overall model architecture and data generation were easy to understand, we encountered the optimization to be especially tricky to perform. In particular, finding a good balance between the reconstruction loss and KL divergence loss was challenging. We implemented GECO [12] to dynamically adapt the Lagrange multiplier but it proved to be surprisingly brittle to its hyper-parameters, resulting in very unstable behavior. We were unable to identify the cause of the problem and ended up training with simpler techniques such as using a fixed Lagrange multiplier as presented in [5].

**Communication with original authors**

We exchanged around 6 emails with doubts and answers with the original authors.

## 1 Introduction

Consider an isolated physical system with multiple bodies interacting with each other. Let $q \in \mathbb{R}^n$ be the vector of their positions, and $p \in \mathbb{R}^n$ the vector of their momenta. The Hamiltonian formalism [3] states that there exists a function $\mathcal{H} : (q, p) \in \mathbb{R}^{n+n} \to \mathbb{R}$ representing the energy of the system which relates $q$ and $p$ as:

$$\frac{\partial q}{\partial t} = \frac{\partial \mathcal{H}}{\partial p}, \qquad \frac{\partial p}{\partial t} = -\frac{\partial \mathcal{H}}{\partial q} \tag{1}$$

In this work $\mathcal{H}$ is modeled with an artificial neural network and property 1 is exploited to get the temporal derivatives of both $q$ and $p$. One can then use a numerical integrator (see Section 4.1) to solve the ODE and infer the system evolution both forward and backward in time given some initial conditions (see Figure 2). These initial conditions are inferred from a natural image sequence of the system evolution (see Figure 1). The authors propose a generative approach to learn low-dimensional representations of the positions and momenta $(q_0, p_0)$. This allows us to sample new initial conditions and unroll previously unseen system evolutions according to the learned Hamiltonian dynamics.

## 2 Scope of reproducibility

The main claim of the paper is that the proposed architecture is able to "learn the Hamiltonian dynamics of simple physical systems from high-dimensional observations without restrictive domain assumptions". This means that the architecture is capable of learning an abstract position and momentum in latent space from RGB images. Then, with the help of an integrator, the architecture will be capable of reconstructing the system evolution. Modifying the integrator time-step will result in a slow-motion or fast-forward evolution. Moreover, the architecture can generate previously unseen system evolutions through sampling. Briefly, we will evaluate the following claims:

- The architecture reconstructs RGB frames of a physical system evolution with an error comparable to [14].
- The architecture can generate new samples qualitatively similar to the originals.
- The timescale of the predicted evolution can be tuned as an integrator parameter without significant degradation of the resulting video sequence.

## 3 Methodology

To date (Jan 1st 2021), authors did not release their code. Therefore, we fully re-implement the Hamiltonian architecture, the integrators, and the simulated environments. To further evaluate the system, we implement two additional environments and one additional integrator. We developed our implementation in Python3 using PyTorch [11] machine learning library for the Hamiltonian architecture and the Scipy [1] ODE solver for the simulated environments, as well as OpenCV [2] for image manipulation. Our code can be found in this repository[1]. We run most of the experiments using an NVIDIA GeForce RTX 2080Ti and some on an NVIDIA GTX 970.

### 3.1 Hamiltonian Generative Network (HGN)

The HGN [14] architecture can be split into two high-level components. The first (Figure 1) reads the initial $k + 1$ frames of an environment rollout and extracts the abstract positions and momenta $(q_k, p_k)$ correspondent to the $k$-th step. Second, a recurrent model takes $(q_k, p_k)$ as first input and performs integration steps of a fixed $\Delta t$, predicting the evolution of the system in terms of abstract positions and momenta [2]. For each step, the abstract position is decoded into an RGB image. As figures 1, 2 depict, this model is composed by four main networks:

- **Encoder**: Parametrized by: $\phi$. 8-layer 64-filter Conv2D network with ReLU activations that takes a sampled video rollout from the environment and outputs the mean and variance of the encoder distribution $q_\phi(z)$ parametrized as a diagonal Gaussian with prior $p(z) = \mathcal{N}(0, \mathbb{I})$. The latent variable $z$ is sampled from $q_\phi$ with the reparametrization trick [9]. The input of this layer is constructed by concatenating all the rollout frames in the channel axis. Therefore, if working with RGB images, the input has shape: $H \times W \times 3 \cdot N$. Where $H, W, N$ are Height, Width, and Number of frames, respectively.

---

[1]https://github.com/CampusAI/Hamiltonian-Generative-Networks
[2]In addition, we test how the network performs when trained as an autoencoder, ie: fit the complete sequence and reconstruct it. (Section 4)

- **Transformer**: Parametrized by: $\psi$. Takes in the sampled latent variable $\boldsymbol{z}$ and transforms it into a lower-dimensional initial state $\boldsymbol{s}_k = (\boldsymbol{q}_k, \boldsymbol{p}_k)$, by applying 3 Conv2D layers with ReLU activations, stride 2, and 64 filters.

- **Hamiltonian**: Parametrized by: $\gamma$. It is a 6-layer 64-filter Conv2D network with SoftPlus activations which takes in the abstract positions and momenta $(\boldsymbol{q}_t, \boldsymbol{p}_t)$ and outputs the energy of the system $e_t \in \mathbb{R}$. This network is used by the integrator (Section 4.1) to compute the system state at the next time-step $(\boldsymbol{q}_{t+1}, \boldsymbol{p}_{t+1})$ exploiting Eq. 1. Since Eq. 1 involves partial derivatives of $\mathcal{H}$ w.r.t. $\boldsymbol{q}$ and $\boldsymbol{p}$, the training process involves second-order derivatives of the Hamiltonian network weights. For this reason, SoftPlus activations are used instead of ReLU.

- **Decoder**: Parametrized by: $\theta$. 3-residual block upsampling Conv2D network (as in [7]) which converts the abstract position $\boldsymbol{q}_t$ into an image close to the source domain.

Given an input sequence: $(\boldsymbol{x}_0, ..., \boldsymbol{x}_T)$ and a value $k + 1$ of input-length, the loss function [3] to optimize is:

$$\mathcal{L}(\phi, \psi, \gamma, \theta; \boldsymbol{x}_0, ..., \boldsymbol{x}_T) = \frac{1}{T+1-k} \sum_{t=k}^{T} \Big( E_{q_\phi(\boldsymbol{z}|\boldsymbol{x}_0,..\boldsymbol{x}_k)} \big[ \log p_{\psi,\gamma,\theta}(\boldsymbol{x}_t \mid \boldsymbol{q}_t) \big] \Big) - \Lambda \cdot KL\big(q_\phi(\boldsymbol{z}) \,\|\, p(\boldsymbol{z})\big) \quad (2)$$

Notice that the loss is the combination of two terms: first, the error coming from the reconstruction of the images, and second, a term which forces the latent distribution $q_\phi$ to be close to a standard Gaussian. It is interesting to see that there is no conditioning over the behavior of latent positions and momenta during the rollout. The architecture connections are enough to force $\boldsymbol{q}_k$ to encode the position information and $\boldsymbol{p}_k$ the momenta information at timestep $k$.

We use the same optimizer as in [14]: Adam [8] with a constant learning rate of $lr = 1.5e-4$ with the GECO algorithm presented in [12] to adapt the Lagrange multiplier $\Lambda$ during training. This Lagrange multiplier is dynamically updated according to an exponential moving average proportional to the reconstruction error of the assessed minibatch. The main parameters controlling the Lagrange multiplier are the exponential moving average constant $\alpha$, the initial Lagrange multiplier, and a parameter to control its growth $\lambda$. The authors did not include the values used in the paper, so we performed a grid search to find the most adequate ones for each environment (see Section 6). In addition, we trained a version of the model with a fixed Lagrange multiplier.

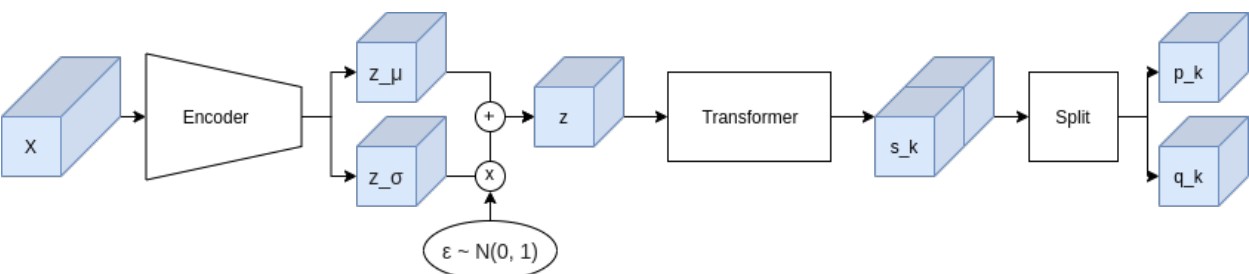

Figure 1: HGN network architecture to find the final abstract position and momentum $(\boldsymbol{q}_k, \boldsymbol{p}_k)$ from the input sequence. Tensors are represented in blue and operations in black. The encoder takes as input a sequence of $k + 1$ frames concatenated along channels and samples the latent variable $\boldsymbol{z} \sim q_\phi(\boldsymbol{z}|\boldsymbol{x}_0, ..., \boldsymbol{x}_k)$ with the reparametrization trick. The transformer network converts $\boldsymbol{z}$ into the state $\boldsymbol{s}_k = (\boldsymbol{q}_k, \boldsymbol{p}_k)$ from which the system evolution will be predicted.

## 3.2 Integrator Modelling

Since the Hamiltonian network always requires backpropagation, which is an expensive operation, we compare it against a baseline network that does not require backpropagation at evaluation time. We test an architecture almost identical to the HGN, but where the Hamiltonian Network is replaced by a CNN that directly computes $\Delta q$ and $\Delta p$ from $q_t$ and $p_t$. Integration is then performed as an Euler step: $q_{t+1} = q_t + \Delta t \Delta q$ and $p_{t+1} = p_t + \Delta t \Delta p$. In this architecture, therefore, we do not learn Hamiltonian-like dynamics anymore, but we directly learn the system dynamics in the abstract space. This approach achieves a similar reconstruction loss than HGN[14]. Results are presented in the additional experiments section. 4.1.

---

[3]The formulation of the loss in Eq. 2 particularly w.r.t. the distribution $q_\phi$ is different from that of the paper[14] where it was written as $q_\phi(\boldsymbol{z}|\boldsymbol{x}_0, ..., \boldsymbol{x}_T)$, which initially led us to think that the encoder had access to the whole rollout. Discussion with the authors clarified that the encoder reads only the first $k$ frames. Therefore, we decided to slightly modify the loss notation in order to avoid confusion. Still, we show results with both approaches to get a more complete idea of the differences.

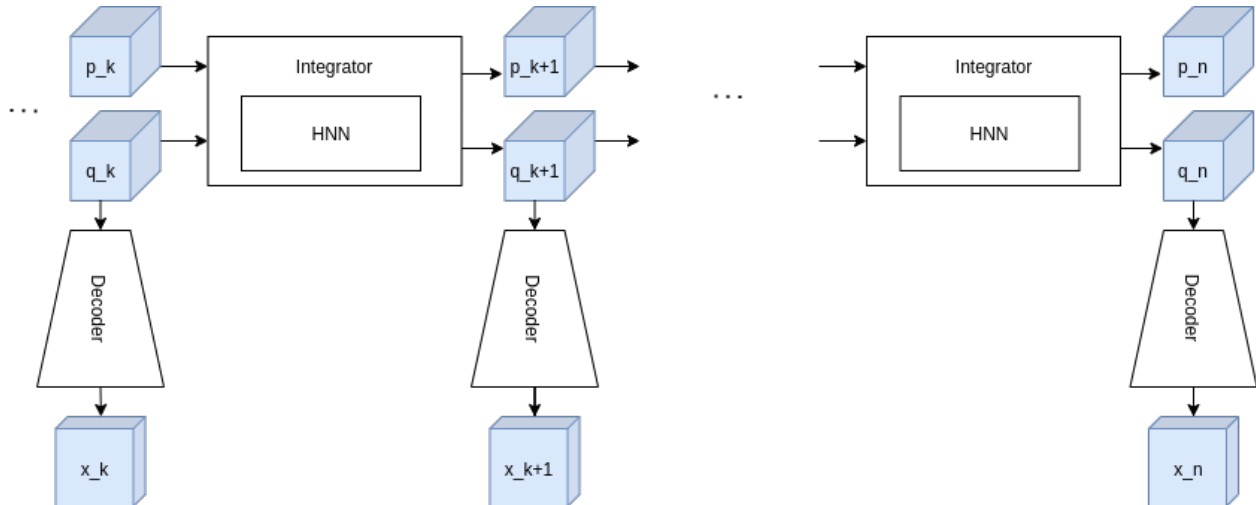

Figure 2: Recurrent part of the HGN architecture. Blue cubes represent tensors. The integrator takes the position and momentum for each time-step, computes $\mathcal{H}(\boldsymbol{q}_t, \boldsymbol{p}_t)$ and computes the abstract state in the next time-step $\boldsymbol{s}_{t+1} = (\boldsymbol{q}_{t+1}, \boldsymbol{p}_{k+1})$ for $t \geq k$ exploiting the Hamiltonian equations of 1. The decoder takes the abstract position $\boldsymbol{q}_t$ and decodes it into the original image $\boldsymbol{x}_t$.

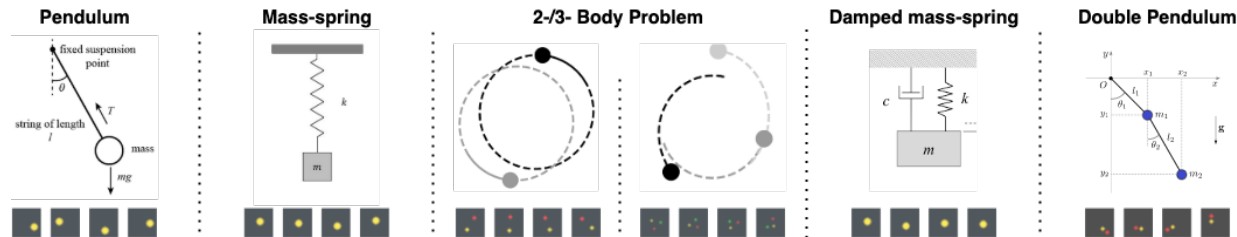

Figure 3: Representation and samples from the different physical systems considered in our experiments. Notice that differing from [14], we also consider a damped mass-spring system and a double pendulum.

## 3.3 Datasets

The datasets considered by the original authors consist of observations of the time evolution of four physical systems: mass-spring, simple pendulum, and two-/three-body systems [14]. Since the datasets are not available to us, we re-implement them following as closely as possible the information provided in the paper and by the authors. Moreover, we introduce two new physical systems to experiment with: damped harmonic oscillator and double pendulum (see Figure 3).

The procedure for data generation is analogous to the one used by [4]. Given a physical system, we first randomly sample an initial state $(\boldsymbol{q}_0, \boldsymbol{p}_0)$ in the phase space and generate a 30 step rollout following the Hamiltonian dynamics. Once the trajectory is obtained, we add Gaussian noise with standard deviation $\sigma = 0.1$ to each phase-space coordinate at each step and render 32x32 image observations. Objects in the systems are represented as circles and we use different colors to represent different objects. We generate 50000 train samples and 10000 test samples for each physical system. To sample the initial conditions $(\boldsymbol{q}_0, \boldsymbol{p}_0)$, we first sample the total energy denoted as a radius $\boldsymbol{r}$ in phase space and then $(\boldsymbol{q}_0, \boldsymbol{p}_0)$ are sampled uniformly on the circle of radius $\boldsymbol{r}$. Note that here $\boldsymbol{q}$ and $\boldsymbol{p}$ represent the actual positions and momenta vectors of the bodies in the system. These are only used to generate the sequence of images and are not made available to the HGN architecture. The trajectories for each environment are computed using the ground-truth Hamiltonian dynamics and SciPy ODE solver [1].

**Mass-spring.** Assuming no friction, the Hamiltonian of a mass-spring system is $\mathcal{H} = \frac{p^2}{2m} + \frac{1}{2}kq^2$, where $m$ is the object's mass and $k$ is the spring's elastic constant. We generate our data considering $m = 0.5$, $k = 2$ and $r \sim \mathbb{U}(0.1, 1.0)$.

| MODEL | MASS-SPRING | | PENDULUM | | TWO-BODY | | THREE-BODY | |
|---|---|---|---|---|---|---|---|---|
| | TRAIN | TEST | TRAIN | TEST | TRAIN | TEST | TRAIN | TEST |
| Orig. HGN (EULER) [14] | $3.67 \pm 1.09$ | $6.2 \pm 2.69$ | $5.43 \pm 2.53$ | $10.93 \pm 4.32$ | $6.62 \pm 3.93$ | $15.06 \pm 7.01$ | $7.51 \pm 3.49$ | $9.4 \pm 3.92$ |
| Orig. HGN (DETERM) [14] | $0.23 \pm 0.23$ | $3.07 \pm 1.06$ | $0.79 \pm 1.24$ | $10.68 \pm 3.19$ | $2.34 \pm 2.3$ | $14.47 \pm 5.24$ | $4.1 \pm 2.05$ | $5.17 \pm 1.96$ |
| Orig. HGN (LEAPFROG) [14] | $3.84 \pm 1.07$ | $6.23 \pm 2.03$ | $4.9 \pm 1.86$ | $11.72 \pm 4.14$ | $6.36 \pm 3.29$ | $16.47 \pm 7.15$ | $7.88 \pm 3.55$ | $9.8 \pm 3.72$ |
| HGN (EULER) ours | $9.05 \pm 0.02$ | $9.06 \pm 0.05$ | $17.79 \pm 0.06$ | $17.86 \pm 0.13$ | $3.84 \pm 0.01$ | $3.85 \pm 0.02$ | $1.99 \pm 0.01$ | $1.99 \pm 0.01$ |
| HGN (DETERM) ours | $7.10 \pm 0.01$ | $7.10 \pm 0.03$ | $14.11 \pm 0.05$ | $14.14 \pm 0.12$ | $3.92 \pm 0.02$ | $3.93 \pm 0.02$ | $4.14 \pm 0.01$ | $4.13 \pm 0.02$ |
| HGN (LEAPFROG) ours | $7.11 \pm 0.01$ | $7.12 \pm 0.03$ | $14.89 \pm 0.05$ | $14.97 \pm 0.1$ | $3.36 \pm 0.02$ | $3.36 \pm 0.02$ | $8.81 \pm 0.01$ | $8.81 \pm 0.01$ |
| HGN (EULER) ours *5-frame inference* | $42.09 \pm 0.14$ | $41.98 \pm 0.32$ | $47.06 \pm 0.17$ | $47.03 \pm 0.39$ | $6.46 \pm 0.03$ | $6.52 \pm 0.06$ | $8.18 \pm 0.01$ | $8.17 \pm 0.01$ |
| HGN (DETERM) ours *5-frame inference* | $13.00 \pm 0.05$ | $13.04 \pm 0.11$ | $45.06 \pm 0.19$ | $44.89 \pm 0.42$ | $10.95 \pm 0.02$ | $10.97 \pm 0.05$ | $3.72 \pm 0.01$ | $3.72 \pm 0.02$ |
| HGN (LEAPFROG) ours *5-frame inference* | $12.15 \pm 0.05$ | $12.21 \pm 0.11$ | $44.29 \pm 0.19$ | $44.12 \pm 0.42$ | $6.28 \pm 0.03$ | $6.33 \pm 0.06$ | $3.35 \pm 0.01$ | $3.35 \pm 0.02$ |

Table 1: Average pixel MSE of the reconstruction of a 30-frame rollout sequence on the test and train datasets of the four physical systems presented by [14]. All the values are multiplied by $10^4$. We show our results (second and third group) along with the ones reported by the original authors (first group). In the second group, we train to reconstruct the whole inputted sequence (as an autoencoder) and in the third group, we train by inputting only the first 5 frames.

**Pendulum.** An ideal pendulum is modelled by the Hamiltonian $\mathcal{H} = \frac{p^2}{2ml^2} + 2mgl(1 - \cos q)$, where $l$ is the length of the pendulum and $g$ is the gravity acceleration. The data is generated considering $m = 0.5$, $l = 1$, $g = 3$ and $r \sim \mathbb{U}(1.3, 2.3)$.

**Two-/three- body problem.** The n-body problem considers the gravitational interaction between $n$ bodies in space. Its Hamiltonian is $\mathcal{H} = \sum_i^n \frac{||\mathbf{p}_i||^2}{2m_i} - \sum_{i \neq j}^n \frac{gm_im_j}{||\mathbf{q}_i - \mathbf{q}_j||}$, where $m_i$ corresponds to the mass of object $i$. In this dataset, we set $\{m_i = 1\}_{i=1}^n$ and $g = 1$. For the two-body problem, we modify the observation noise to $\sigma = 0.05$ and set $r \sim \mathbb{U}(0.5, 1.5)$. When considering three bodies, we set $\sigma = 0.2$ and $r \sim \mathbb{U}(0.9, 1.2)$.

**Dobule pendulum** The double pendulum consists of a system where we attach a simple pendulum to the end of another simple pendulum. For simplicity, we conider both simple pendulums with identical properties (equal mass and length). The Hamiltonian of this system is $\mathcal{H} = \frac{1}{2ml^2} \frac{p_1^2 + p_2^2 + 2p_1p_2 cos(q_1 - q_2)}{1 + sin^2(q_1 - q_2)} + mgl\left(3 - 2\cos q_1 - \cos q_2\right)$, where $\{q_1, p_1\}$ and $\{q_2, p_2\}$ refer to the phase state of the first and second pendulum respectively. Our data is generated by setting $m = 1$, $l = 1$, $g = 3$ and $r \sim \mathbb{U}(0.5, 1.3)$. In this scenario we consider a very low intense source of noise $\sigma = 0.05$.

**Damped oscillator** The damped mass-spring system is obtained by considering a dissipative term in the equations of motion of the ideal mass-spring system. For such systems, one can obtain its dynamics using the Caldirola-Kanai Hamiltonian $\mathcal{H} = e^{\gamma t}\left(\frac{p^2}{2m} + \frac{1}{2}kq^2\right)$ [13], where $\gamma$ is the damping factor of the oscillator. In our experiments, we consider an underdamped harmonic oscillator and set $\gamma = 0.3$, $m = 0.5$, $k = 2$, $r \sim \mathbb{U}(0.75, 1.4)$ and $\sigma = 0.1$.

### 3.4 Hyperparameters

We set the same hyperparameters for all experiments as the original paper [14] except for GECO parameters, which were not included. Thus, we perform a grid search on each environment to find the most adequate ones (see Section 4.1).

### 3.5 Computational requirements

A standard training of 50K train samples using the Leapfrog integrator takes around 4 hours on an RTX 2080T GPU and requires around 1910MB.

## 4 Results

We first test whether the HGN [14] can learn the dynamics of the four presented physical systems by measuring the average mean squared error (MSE) of the pixel reconstructions of each predicted frame. Furthermore, we test the original HGN architecture along with different modifications: a version trained with Euler integration rather than Leapfrog integration (HGN Euler), and a version that does not include sampling from the posterior $q_\phi(z|x_0...x_T)$ (HGN determ). Since we could not find suitable GECO[12] hyperparameters, we use a fixed Lagrange multiplier[5] in all the experiments.

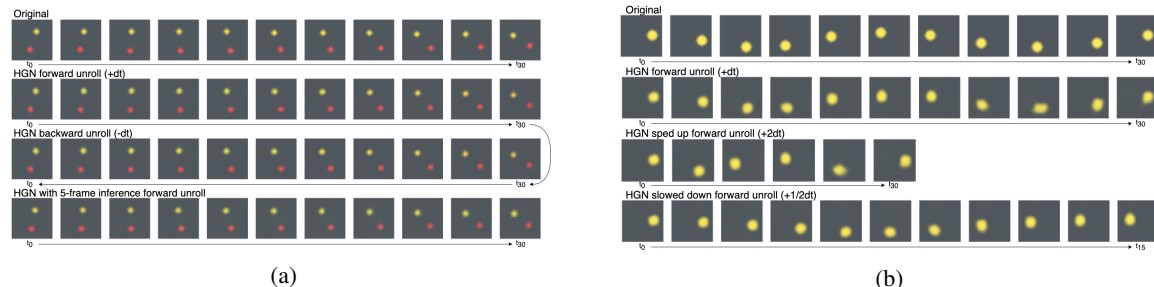

(a)

(b)

Figure 4: (a) Reconstruction of a sequence of the 2-body system along with a backward unroll of the data from the final state, and a forward rollout of the HGN trained using state inference from the first 5 frames. (b) Reconstruction of a sequence of the pendulum system along with a sped up and a slowed down forward rollout.

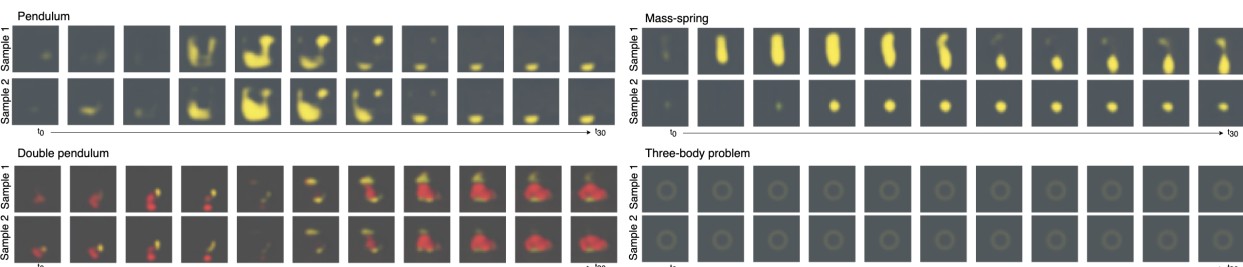

Figure 5: Examples of sample rollouts from the latent space for different physical systems.

Table 1 shows the results of the experiments described previously along with the results of the original authors. As it can be seen, we achieve average pixel reconstruction errors that are similar (30% avg absolute error w.r.t. the reported values on the test set using Leapfrog integrator) to the ones reported in the original paper when reconstructing the same sequence that is inputted (we call this version *autoencode*). However, when attempting to train to reconstruct a rollout given only the first 5 frames our model performs poorly, with 107% average absolute error on the test set, using Leapfrog integrator.

In Figure 4, we show some qualitative examples of the reconstructions obtained by the full version of HGN. The model can reconstruct the samples and its rollouts can be reversed in time, sped up, or slowed down by changing the value of the time step used in the integrator. Since the HGN is designed as a generative model, we can sample from the latent space to produce initial conditions and perform their time evolution. We show some rollouts obtained this way in figure 5. We observe that our model is only able to generate plausible and diverse samples in the mass-spring dataset. This behavior is different than the one shown by [14] and might be caused by different hyper-parameter configurations in the training procedure or some implementation mistake.

We achieve slightly larger MSE in the autoencode version and significantly larger in the 5-frame inference problem on both the mass-spring and pendulum. The latter presents roughly double MSE probably because of a wider span of movement. In general, these two environments show worse results in comparison to two/three-bodies. For these last cases, our implementation using the *autoencode* setting outperforms the original HGN [14], and when using the *5-frame inference* the results are similar. As we can see, these two environments show much less average pixel MSE compared to the first ones (almost one order of magnitude). We believe this may be due to the differences when rendering the instances of each dataset. The elements appearing in mass-spring and pendulum (represented by a large yellow ball) are larger than the ones present in the two/three bodies (two/three small coloured balls). Because of this, it would be reasonable to assume that localization errors are more penalized in the first two environments, since the total difference in areas is larger. Furthermore, the dynamics representing mass-spring and pendulum show faster movements in comparison to two/three-bodies, resulting in being harder to represent with our HGN. Consequently, we hypothesise the following: larger elements and faster dynamics, produces higher average MSE on our model regardless of the difficulty of the environment physics. However, this is not the case for the original author's results, who seem to struggle more on the two/three-bodies. Surprisingly, it seems that our hyperparameter and architecture choices led to poorer reconstruction capabilities (higher MSE) but learning better physics (qualitatively more realistic movements).

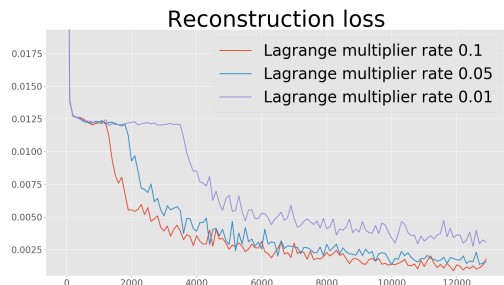
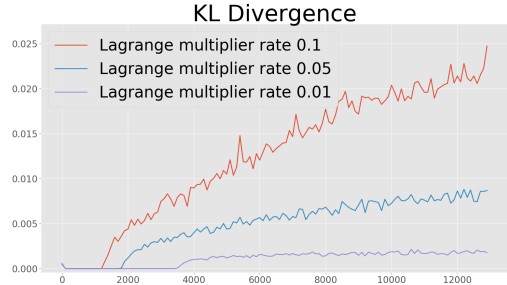

Figure 6: Reconstruction loss and KL divergence for different GECO parameters in the Pendulum environment.

|  | EULER | RUNGE-KUTA 4 | LEAPFROG | YOSHIDA |
|---|---|---|---|---|
| pixel MSE | $17.86 \pm 0.13$ | $76.88 \pm 0.08$ | $14.97 \pm 0.10$ | $14.70 \pm 0.10$ |
| $\mathcal{H}$ std | 3.81 | 0 | 1961.93 | 1893.05 |
| reconstr. time (s) | 0.32 | 1.89 | 0.96 | 1.61 |

Table 2: Comparison between four different integrators used to perform the time evolution in the HGN. The results are measured on the simple pendulum test set. The pixel MSE values have been multiplied by $10^4$.

## 4.1 Additional experiments

**GECO parameter search** The paper does not provide the values of GECO [12] used. In GECO, the Lagrangian multiplier is optimized at each step with a rate $\gamma$. Figure 6 shows the behavior of GECO for $\gamma \in \{0.1, 0.05, 0.01\}$ in terms of reconstruction loss and KL divergence. Higher values of $\beta$ give a better reconstruction loss but greatly increase the KL divergence. However, we found that hyperparameters were not consistent among different environments and integrators. For this reason, we do not use GECO in our experiments.

**Integrators** Performing the integration step is key to generate the time evolution of a rollout given the initial state. In the HGN paper [14] the system is tested using Euler and Leapfrog integration. We wonder if using higher order integration methods might boost the performance of the rollout generation process. Therefore, we implement and test the HGN architecture with two additional numerical integration methods: the Runge-Kutta's 4th-order integrator [6] and the 4th-order Leapfrog integrator (Yoshida's algorithm [15]). Table 2 shows a comparison of all four integrators on the Pendulum dataset. Both Leapfrog and Yoshida are *symplectic* integrators: they guarantee to preserve the special form of the Hamiltonian over time [10].

Table 2 shows the average pixel MSE, the averaged standard deviation of the output of the Hamiltonian network during testing, and the reconstruction time of a single batch (`batch = 16`) using the different integration methods that we have described previously. The model has been trained on the simple pendulum dataset. As we can see, the reconstruction time increases when using higher-order integration methods, since they require more integration steps. In general, we see that Euler integration offers a fast and sufficiently reliable reconstruction of the rollouts. Moreover, we observe that the fourth-order symplectic integrator (Yoshida) achieves the best performance. Surprisingly, the symplectic integration methods show more variance in the output of the Hamiltonian networks throughout a single rollout. This behavior is unexpected since using a symplectic integration method should ideally keep the value of the Hamiltonian invariant. We conclude that more experiments need to be performed to guarantee that the implementation of both Leapfrog and Yoshida integration methods are faithful to their formulation.

**Integrator modelling** We train the modified architecture of Section 3.2 on the Pendulum dataset for 5 epochs. The architecture is the same as HGN, but the Hamiltonian Network now outputs $\Delta q$ and $\Delta p$. The average MSE error over the whole Pendulum dataset is $1.485 \times 10^{-3}$, while in the test set it is $1.493 \times 10^{-3}$, which are both very close ($\sim \pm 2\%$) to those of autoencoding HGN (see Table 1). The modified architecture is still capable of performing forward slow-motion rollouts by modifying $\Delta t$. We set $\Delta t' = \frac{\Delta t}{2}$ and we compute the average MSE of the slow-motion reconstruction over 100 rollouts. The modified architecture achieved an error of $8x10^{-4}$, while the standard HGN achieved $9x10^{-4}$. Note that reconstruction losses are smaller for slow-motion as the images change less between timesteps.

**Extra environments** Apart from the four physical systems presented by [14] we test our re-implementation of the HGN with physical systems that do not have a simple Hamiltonian expression. As described previously, these are

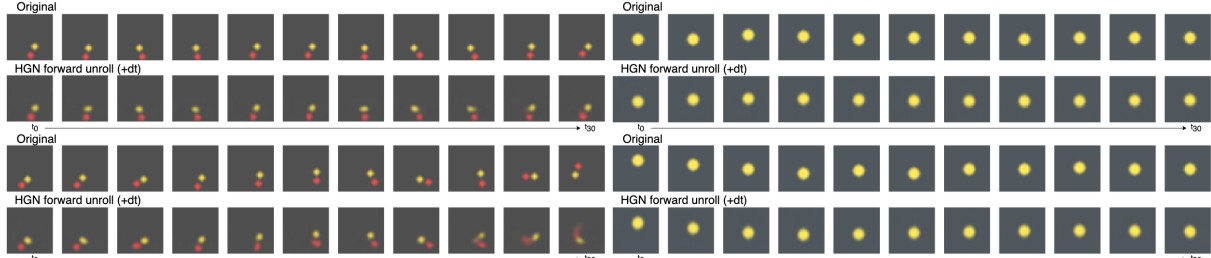

Figure 7: Examples of reconstructions of the double pendulum (left) and the damped harmonic oscillator (right).

the damped harmonic oscillator and the double pendulum. On one hand, we are interested in a damped system since it introduces a dissipative term to the equations of motion; a feature that differs from the previous systems. On the other hand, the double pendulum is modelled by a non separable Hamiltonian: $\mathcal{H}(\mathbf{q}, \mathbf{p}) \neq K(\mathbf{p}) + V(\mathbf{q})$ as described previously. In figure 7 we show some visual examples of the reconstructions provided by the HGN trained on the two systems. As we can see, HGN is able to reconstruct the damped oscillator with high reliability. Regarding the double pendulum, we observe that the model reconstructs well small oscillations, but fails when the trajectory is too chaotic as expected. The average pixel MSE of the reconstructions of the damped oscillator and the double pendulum are $6.39 \cdot 10^{-4}$ and $6.91 \cdot 10^{-4}$ respectively. The HGN is able to provide better reconstructions for these systems in comparison to the mass-spring and pendulum systems.

# 5 Discussion

We were able to implement and train an Hamiltonian Generative Network with similar reconstruction performance of the ones of the original paper (30% average absolute relative error wrt to their reported values when treating it as an autoencoder). These results show that the network is capable of exploiting the Hamiltonian equations to learn dynamics of a physical system from RGB images. However, the value of the resulting Hamiltonian does not remain constant throughout the system evolution. This means that the network is learning something that is different from the Hamiltonian equations described in Section 3.3.

To make the variational sampling work, we tried performing a grid search on the Geco[12] hyperparameters and using a fixed Lagrange multiplier as in [5]. However, despite our best efforts, the samples produced by the variational model have very poor quality. This is generally due to the difficulty in minimizing both KL divergence and reconstruction loss.

We believe that further experiments are needed to understand better the behavior of the system and to improve it. Future work could include further testing on each network architecture, probably smaller networks would also be able to encode the needed information. Another next step is to try the approach on more challenging (and realistic-looking) environments. In addition, it would be interesting to tackle the transfer learning capabilities of such architecture between different environments. How re-usable each network is? How much faster the system is able to learn the new dynamics? Finally, another field which could benefit from this research is model-based reinforcement learning. A generative approach from which to sample example rollouts could be very useful for training agents without the need of directly interacting with the environment.

## 5.1 What was easy

Once we implemented the code it resulted quite easy to perform multiple experiments on different environments, architectures and hyper-parameters due to the code's modularity and flexibility. We can define the the previously mentioned experiments and most common testing behaviors from a set of yaml files which can then be modified from command-line arguments. While this required extra planning and work at the beginning it really payed off when debugging and evaluating in later stages.

## 5.2 What was difficult

The main challenge we encountered is finding the correct tools to debug a model composed of so many interconnected networks. The fact that it has a variational component with a dynamic Lagrange multiplier term makes it especially tricky to train. Furthermore, no public implementation existed and some details and parameters were missing in the original paper leading to some necessary assumptions or parameter searches.

## 5.3 Communication with original authors

We first tried to understand and re-implement the code by ourselves. Nevertheless, at some point we had gathered a significant set of doubts and we decided to email them to the original authors, which they answered with great detail. From that point onwards, we sent a couple more set of doubts, also receiving answers.

Most of our doubts were about network architecture clarifications (either of unclear or missing descriptions from the original paper), and loss function evaluation. Furthermore, they provided us with some of their environment images so we could more easily make our environments as similar as possible.

## 5.4 Improving reproducibility

Having worked in re-implementing the whole original work, we feel it is important to share our experience as well as providing a recommendation on how it could be made more easily reproducible. First, having the environments data or code to generate it available online would save the effort and, most importantly, it would constitute a baseline against which to compare future work. Secondly, publishing all the hyperparameters and more details of the networks architecture would make the whole work much easier to reproduce and require less training attempts, especially for what concerns GECO.

# Acknowledgements

We thank Stathi Fotiadis for voluntarily contributing with a GECO [12] implementation draft to the public repo and his useful feedback on code structuring. We thank the KTH Robotics, Perception, and Learning (RPL) Lab for the computational resources provided to us. In addition, we would like to thank the original authors for providing further details on the implementation.

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
