# OpenReview forum: "Re-Hamiltonian Generative Networks"
_ML_Reproducibility_Challenge/2020 — RC2020_

### Official Review · AnonReviewer2 · 2021-02-18
**Good Reproducibility Project**

**Rating:** 7
**Confidence:** 3

**Review:**

The authors have chosen to reproduce results related to Re-Hamiltonian Generative Networks. The code was built from scratch in Pytorch, based on the description of the original paper. Results are consistent with the original ones for similar testbenches, but show sub-optimal behavior on new data. It is also found that results are highly dependent on hyper-parameter tuning.

The reproducibility report is very well written, the problem is first formulated, the methodology is clearly presented, and the results are well described. The authors also indicate communications with those of the original paper, which is good to see.

**Familiar With The Original Paper:**

I have not read the original paper

**Reproducibility Summary:**

Report has summary

---

### Official Review · AnonReviewer3 · 2021-02-26
**Good reimplementation with some differences due to unknown Lagrage multiplier**

**Rating:** 7
**Confidence:** 4

**Review:**

The report clearly summarizes the problem statement of the original paper "Hamiltonian Generative Networks" (HGN) as scope of reproducibility: learn a Hamiltonian dynamics from a image sequence. As no source code was available, the authors reimplemented the program for the experiments from scratch based on the description in the original paper and questions about implementation details answered by the original authors.

The code is available on GitHub as a Python program based on Pytorch. It is properly documented and cleanly written. All dynamical systems from the original paper have been implemented and the experiments replicated. Results are comparable with the original paper in autoencoder mode, where training happens with the full input sequence. But using only the first five frames for training the HGN leads to significantly worse results for two systems (mass-spring and pendulum), but not for the other two (two-body and three-body). This difference is not discussed, only the average over all four systems. I suspect that this is caused by the choice of the Lagrage multiplier, as the authors did not obtain sufficient information about the automatic optimization of this hyperparameter.

Additionally, the authors present successful results using other integrators and extra dynamical systems. They also show that the calculation of the derivatives for the Hamiltonian equations of motion by backpropagation can be replaced by a network which learns them directly.

The discussion clearly indicates the state of reproducibility and summarizes the easy and difficult parts of that task. There were no explicit recommendations for better reproducibility, but the authors describe, what information missing in the original paper could have been helpful.

The reproducibility report is well written. I recommend to add a discussion about the differences of reproducibility between the dynamical systems and to make an explicit recommendation to the original authors for improving reproducibility.

**Familiar With The Original Paper:**

I have read the original paper

**Reproducibility Summary:**

Report has summary

---

### Official Review · AnonReviewer1 · 2021-03-03
**Significant effort on an important yet opaque paper**

**Rating:** 7
**Confidence:** 4

**Review:**

Arguably, learning how physical systems work is one of the key unsolved problems in machine learning. The present report investigates an attempt to solve such a problem with significant results but one that doesn't reveal a lot about it's inner workings. The author(s) do a great job in reproducing, replicating, and presenting the challenges. The report also communicates the core problem very well and their contact with the authors of the original paper is transparent.

What would improve the paper:
- testing hyperparameter sensitivity would be useful. using exactly as the ones indicated is good for replication but not enough to understand the scope of the sensitivity of the model.
- testing extra environments is a great addition, and I think it would also benefit a lot from understanding failing cases a bit deeper.
It seems the authors are aware of possible improvements and some of them can be standalone work by themselves.

====

trivial typos:
line 45 an —> and
line 214 or —> our

**Familiar With The Original Paper:**

I have read the original paper

**Reproducibility Summary:**

Report has summary

---

### Decision · Program_Chairs · 2021-03-31

**Decision:**

Accept

**Comment:**

Strong reproduciblity report; building the code from scratch is definitely an important contribution, as well as establishing that the results are dependent on hyperparameter tuning.